# BAYWRF: a high-resolution present-day climatological atmospheric dataset for Bavaria

Emily Collier[1] and Thomas Mölg[1]

[1]Climate System Research Group, Institute of Geography, Friedrich-Alexander University Erlangen-Nürnberg (FAU), Erlangen, Germany

*Correspondence to*: Emily Collier (emily.collier@fau.de)

**Abstract.**

Climate impact assessments require information about climate change at regional and ideally local scales. In dendroecological studies, this information has traditionally been obtained using statistical methods, which preclude the linkage of local climate changes to large-scale drivers in a process-based way. As part of recent efforts to investigate the impact of climate change on forest ecosystems in Bavaria, Germany, we developed a high-resolution atmospheric modelling dataset, BAYWRF, for this region over the thirty-year period of September 1987 to August 2018. The atmospheric model employed in this study, WRF, was configured with two nested domains of 7.5- and 1.5-km grid spacing, centred over Bavaria and forced at the outer lateral boundaries by ERA5 reanalysis data. Using an extensive network of observational data, we evaluate: (i) the impact of using grid-analysis nudging for a single-year simulation of the period of September 2017 to August 2018; and (ii) the full BAYWRF dataset generated using nudging. The evaluation shows that the model represents variability in near-surface meteorological conditions generally well, although there are both seasonal and spatial biases in the dataset that interested users should take into account. BAYWRF provides a unique and valuable tool for investigating climate change in Bavaria with high-interdisciplinary relevance. Data from the finest resolution WRF domain are available for download at daily temporal resolution from a public repository at the Open Science Framework (Collier, 2020; https://www.doi.org/10.17605/OSF.IO/AQ58B).

## 1 Introduction

The forcing of climate change in modern times is clearly of global nature, and many important scientific problems can be understood at the global scale as well (e.g., Held and Soden, 2006). Climate impact assessments, however, must also understand the effects at regional and even local scales in order to develop appropriate adaptation and mitigation measures. Although local phenomena such as glaciers, lakes, vegetation patterns, or stream flow show a strong dependence on the large-scale climate dynamics, these proxies experience further variability when the large-scale signal is transferred to their

location (e.g., Mölg et al., 2014). In order to contextualize local changes, there is a need to link local climate to the large-scale climate, ideally in a process-based way.

In dendroclimatological studies, the traditional approach is to compute a calibration function between local or regional tree-ring parameters and climatic variables. Typically, such a statistical relationship would try to utilize local station data (which are generally sparse), gridded observations (which tend to be coarse resolution), or indices of large-scale climate dynamics (which describe coupled atmosphere-ocean modes) as the climatic influence (e.g., Hochreuther et al., 2016). Besides known problems like stationarity (e.g., Frías et al., 2006), statistical approaches also limit the possibilities to explain the influences at the various scales on a process-resolving level. Dynamical downscaling with a full numerical atmospheric model provides a physical answer (Giorgi and Mearns, 1991), yet the disadvantage is the high computational cost. Hence, dynamical downscaling at near-kilometer resolution has traditionally been performed on a case-study basis for weather events (e.g., Gohm et al., 2008). Multi-decadal simulations, on the other hand, were typically limited to resolutions of tens of kilometers (e.g., Di Luca et al., 2016). With the progress of computational resources, dynamical downscaling is becoming a candidate for climate impact studies that require local-scale information, and the first decadal simulations at ~1-km resolution are now available (e.g., Collier et al., 2018). From the resultant model output, impact studies could utilize information about local meteorological conditions at high-spatial and high-temporal resolution, and over long, climatologically relevant temporal periods. Moreover, the physically consistent output would enable to generate the said process understanding of influences across the various climatic scales.

The management of forests is a classical impact study where adaptation and mitigation measures meet the heterogeneous effects of climate change at local scales (e.g., Lindner et al., 2014). With this background, the project BayTreeNet was started recently under the umbrella of the interdisciplinary climatological research network Bayklif (https://www.bayklif.de; last accessed 1 March 2020), and aims to investigate the response of forest ecosystems to current and future climate dynamics across different growth areas in Bavaria, Germany. The project comprises a network of 10 measurement sites where meteorological and dendroecological data will be monitored and used both for research and for public and educational outreach, which are currently in the process of being established. High-temporal (approximately daily) and high-spatial resolution data is a key component of dendroecological impact studies, since the physiological behavior of trees, their structural properties and functional wood anatomy, as well as other important parameters such as wood density and mortality risk are not only influenced by seasonal averages, but also by short-term extreme events and weather anomalies (e.g., Bräuning et al., 2016).

Previous regional climate simulations including Bavaria over continuous multi-decadal periods were performed with model resolutions as high as 5-7 km and up to the year 2009 (e.g., Berg et al., 2013; Warscher et al., 2019). However, to the best of our knowledge, such datasets at the kilometer scale and up to the near present do not yet exist, despite previous research

highlighting the importance of convection-permitting resolution in this region (Fosser et al., 2014). We address this data gap by performing simulations with an atmospheric model, configured with convection-permitting spatial resolution in a nested domain over Bavaria, for the recent climatological period of 1987 to 2018. These data have the potential to find multidisciplinary interest among researchers assessing ecological and human dependencies on the climate for scientific and practical questions.

## 2 Methods

### 2.1 Atmospheric model

The atmospheric simulations were performed using the advanced research version of the Weather Research & Forecasting (WRF) model v. 4.1 (Skamarock and Klemp, 2008) configured with two one-way-nested domains of 7.5- and 1.5-km grid spacing situated over Bavaria (Fig. 1), hereafter referred to as D1 and D2. Terrain data were taken from NASA Shuttle Radar Topographic Mission data re-sampled to 1-km and 500-m grids (Jarvis et al., 2008; https://cgiarcsi.community/data/srtm-90m-digital-elevation-database-v4-1; last accessed 24 May 2020) for D1 and D2, respectively, while land-use data was updated based on the European Space Agency Climate Change Initiative Land Cover data at 300-m spatial resolution (http://maps.elie.ucl.ac.be/CCI/viewer/download.php; last accessed 18 April 2018). The physics and dynamics options used in the simulations are based on several recent convection-permitting applications of WRF by the authors (e.g., Collier et al., 2019) but were not specifically optimized for these domains due to the computational expense of the simulations. The options are summarized in Table 1 and a sample namelist is provided in Appendix A. As no cumulus parameterization was employed in D2, both deep and shallow convection are assumed to be explicitly resolved. We note that no additional urban physics were enabled beyond the default parameterization used by the Noah family of land surface models (Liu et al., 2006) and land-use sub-tiling was not enabled.

Forcing data at the lateral boundary of D1 and bottom boundaries of both domains was taken from the ERA5 reanalysis (Copernicus Climate Change Service (C3S), 2017) at three-hourly temporal resolution. The 30-year simulation was divided into 30 annual simulations that were run continuously from 15 August of year $n$-$1$ to 31 August of year $n$. The first 16 days of each simulation were discarded as spin-up time, retaining data from 1 September of year $n$-$1$ onwards. Atmospheric carbon dioxide ($CO_2$) was updated in WRF for each simulation year using annually and globally averaged concentrations at the surface from the National Oceanic and Atmospheric Administration Earth System Research Laboratory (Tans and Keeling, 2019). Each simulation employed the $CO_2$ concentration of year $n$, which ranged from 351 to 407 ppm between 1988 and 2018. All other parameters and bottom boundary conditions (e.g., vegetation and land use) were held constant for all simulations. Therefore, they do not capture the impact of known land-use changes over the study period (e.g., Fuchs et al., 2013).

Each run required 12 days of wall-time with 320 processors on the Meggie compute cluster at the Erlangen Regional Computing Center, for a total of 2.86 million core hours. The model was compiled using intel 17.0 compilers and run using distributed-memory parallelization. Model output was written at two-hourly intervals, amounting to more than 55 TB of data, in addition to ~30 TB of pre-processing and input files. We selected this write frequency as a compromise between high-temporal resolution and the logistical challenges of storing, analyzing, and disseminating the data.

## 2.2 Evaluation of Forcing Strategy

For the period of 00 UTC 1 September 2017 to 00 UTC 1 September 2018, we compared two simulations with different forcing approaches: one excluding and one including grid-analysis nudging to constrain drift in the large-scale circulation (e.g., Bowden et al., 2013). This period was selected due to the higher availability of observational data closer to present day and because the summer of 2018 contained a record heatwave with drought conditions (Beyer, 2018), permitting evaluation
of an extreme event. We refer to these simulations as WRF_NO_NUDGE and WRF_NUDGE, respectively. For the WRF_NUDGE simulation, analysis nudging was applied in D1 outside of the planetary boundary layer and above the lowest 10 model levels using the default strength ($3.0 \times 10^{-4}$) for temperature and winds and reduced strength ($5.0 \times 10^{-5}$) for the water vapor mixing ratio (e.g., Otte et al., 2012), consistent with a previous decadal application of WRF (Collier et al., 2018). Given the computational expense of each annual simulation, we did not attempt to optimize the nudging coefficients
for our study area and instead evaluate simply whether nudging in this form improves the simulated atmospheric variables or not.

## 2.3 Observational Data

For model evaluation, we used data from the German Weather Service (DWD) Climate Data Center for all stations in Bavaria with hourly temporal resolution available, which provide good spatial coverage of our study area (Table 2; Fig. 2).
To evaluate the forcing approach, we compared the following near-surface atmospheric variables at the highest temporal resolution available in the simulations, which is two-hourly: air temperature and relative humidity at 2 m ($T$ and $RH$), zonal and meridional wind components at 10 m ($U$ and $V$), and surface pressure ($PS$). In addition, we compared with daily total precipitation ($PR$). In our comparison with observations, we excluded measurement sites where the observed terrain height differed from the modelled value by more than 100 m (similar to e.g., Vionnet et al., 2019), corresponding to four sites in
total for all variables except for $PS$ (three) and $PR$ (nine). After this exclusion, the average difference between modelled and observed terrain height at all stations is within ± 8 m for each dataset. We also excluded any days with missing observational data when computing daily statistics. We note that observed precipitation was not corrected for undercatch. We did not evaluate radiation variables, as only sunshine hours are available from the DWD in sufficiently large sample sizes. However, for understanding temperature biases in WRF during summer 2018, we used incoming shortwave radiation from the DWD
Climate Data Center dataset entitled "*Hourly station observations of solar incoming (total/diffuse) and longwave downward radiation for Germany*" (Table 2). In total, there were four sites with both incoming shortwave ($SW$) and $T$ data available in

Bavaria between 1 June and 31 August 2018 whose elevation was represented within ±100 m in D2: Nürnberg (id 3668), Weihenstephan-Dürnast (5404), Würzburg (5705), and Fürstenzell (5856).

For statistical analysis, we computed the mean deviation (MD), mean absolute deviation (MAD), and the coefficient of determination ($R^2$) between station data and data from the closest grid point in D2 without spatial interpolation at two-hourly and, for precipitation, at daily temporal frequency. The MD, also referred to here as the model bias, and the MAD were computed from observation minus model data. For precipitation, only daily totals were evaluated, and the MD and MAD were computed considering only days with non-zero observed precipitation.

    Finally, we also compared night-time land surface temperature (LST) from the MODIS MYD11A1 dataset (Table 2) at 1-km spatial and daily temporal resolution with simulated skin temperature in D2 for the period of 1 June to 31 August 2018. The night view time ranged from 1.2 to 2.8 hours in local solar time, with a domain and time averaged value of 2.2 hours. As WRF data were only available at two-hourly timesteps, we averaged 00 and 02 UTC (01 and 03 local time) data from D2 for
comparison with MODIS. In our comparison, we excluded nights when MODIS had more than 50% missing data over D2, leaving a sample size of 52.

    For evaluating the full simulation, we performed a similar analysis with the aforementioned station datasets for *T*, *RH* and *PREC* (Table 2), however we averaged and summed the data to daily timescales for comparison with BAYWRF. In addition
to comparing with individual stations, we also compared monthly total precipitation in BAYWRF with the gridded dataset REGNIE from the DWD CDC, which is based on interpolated station data and available at 1-km spatial resolution (e.g., Rauthe et al., 2013). For the comparison, REGNIE data were regridded to the WRF grid using patch interpolation and the ESMF regridding toolbox in NCL (https://www.ncl.ucar.edu/Document/Functions/ESMF/ESMF_regrid.shtml; last accessed 10 September 2020) and the centered pattern correlation between the two datasets was computed.

**2.4 Numerical issue in BAYWRF**

    We note that unphysically large sub-surface temperatures were simulated at a number of glacierized grid points, primarily during the months of July to September. Considering all of D2, the daily average number of affected grid cells was 24, compared with 294 glacierized and 122,500 total cells. The maximum number of affected grid points was 274 on 31 August 2017, corresponding to 0.2% of D2. In addition, over the climatological simulation, only one grid point in Bavaria was
affected (J = 71, I = 285; 47.4952°N, 13.6039°E). Surface temperature remained physical, since it is limited at the melting point over glacier surfaces, and soil moisture was unaffected, since it is specified to be fully saturated in glacierized grid cells. No other land-use categories were affected, and adjacent grid points were also unaffected, as the land surface model operates as a column model with no lateral communication. To preclude usage of these data, sub-surface temperature was set to missing where it exceeded the melting point at glacierized grid points in BAYWRF. More information about this

numerical issue is available on the model's GitHub repository (https://github.com/wrf-model/WRF/issues/1185; last accessed 24 May 2020).

## 3 Results & Discussion

### 3.1 Evaluation of forcing approach

Averaged over the evaluation year, both WRF simulations capture the magnitude and variability of sub-diurnal near-surface
meteorological conditions at most sites well (Fig. 3; Table 3). The interquartile range (IQR; range between upper and lower quartile) of MDs is one order of magnitude smaller than the observed standard deviation for all variables. As expected, variability is best captured for $T$ and $PS$, with $R^2$ values that uniformly exceed 0.87 and 0.96, respectively. Those of $RH$ have a larger range but a lower quartile above ~0.55. Compared with these variables, the model shows less skill in simulating sub-diurnal variability in winds, with lower quartiles of $R^2$ for $U$ and $V$ of approximately 0.39 and 0.27, respectively.


Shifting to daily timescales, both simulations represent variability in daily total $PR$ surprisingly well, with the upper quartile of MDs below ~1.25 mm and lower quartiles of $R^2$ exceeding 0.18 and 0.33, depending on the simulation. The MD is positive at the majority of stations, indicating that WRF generally underestimates observed precipitation. The underestimate is likely greater than reported here, since the observations were not corrected for wind-induced undercatch. In addition to
underestimating observed daily precipitation events (total sample size of 35,791 for all stations and record lengths), the simulations also produce false daily precipitation events, the vast majority of which are very small in magnitude (the median value in both WRF simulations is less than 0.1 mm/day). Considering wetter days (precipitation exceeding 1 mm/day; Ban et al., 2014), the number of false events is more than ten times smaller than the number of observed events (sample sizes of 3,096 and 2,249 in WRF_NO_NUDGE and WRF_NUDGE, respectively).


Previous studies evaluating WRF over this region have reported Root Mean Square Deviations (RMSD). For direct comparison, the mean RMSD in WRF_NUDGE for two-hourly $T$ and $RH$ is 2.67°C and 13.7%, respectively, and for daily total precipitation is 5.27 mm. These values are comparable to previous high-resolution applications of WRF over Bavaria (Warscher et al., 2019).


Examination of model biases on a monthly basis reveals further insights into the model performance (Fig. 4). The amplitude of the annual cycle is overpredicted in WRF, indicating that the good average agreement in $T$ results from compensating biases: there is a cold bias in WRF in winter, a well-known issue with the model over snow-covered surfaces (e.g., Tomasi et al., 2017), and a warm bias in summer (Fig. 4a). The latter bias results in an underprediction of $RH$ during this season (Fig.
4b), suggesting that WRF represents absolute humidity more accurately. The summer temperature bias is also more sustained than the winter one, resulting in the long tails (heads) in the distribution of MDs of $T$ ($RH$) in Fig. 3. There is also a general

underprediction of near-surface winds from fall to early winter, as exemplified by the results for $U$ in Fig. 4c and the slight positive skewness of the distribution of MDs for both $U$ and $V$ in Fig. 3, consistent with overly stable atmospheric conditions resulting from the cold bias. Finally, the model tends to overestimate precipitation in early spring and underestimate it in summer and fall. The reported seasonal and mean biases in daily precipitation are consistent with a potential underestimate of deep convection and convective precipitation at 1.5-km grid spacing. Although simulated mean precipitation shows a weak grid dependency below a spacing of ~ 4 km (Langhans et al., 2012), sub-kilometer spatial resolution is required to explicitly resolve the evolution and characteristics of clouds (e.g., Bryan et al., 2003; Craig and Dörnbrack, 2008; Prein et al., 2015).

Figure 5 shows a representative timeseries of $T$ and $SW$ for the station in Nürnberg (3668) in June 2018. The timeseries illustrates that the positive temperature bias in summer 2018 results from two distinct contributions. First, there is an overestimation of daytime maximum $T$, coinciding with an overestimation of $SW$. This relationship is observed both at Nürnberg and at the other three stations for which both datasets are available (Fig. 6a; cf. Sect. 2.2). The overestimation suggests there is an underestimation of either daytime cloudiness or its impact on incoming $SW$ at the surface, likely stemming from the microphysics parameterization. Ban et al. (2014) identified similar processes underlying a warm bias in summer in a convection-permitting decadal simulation over central Europe. Second, there is an overestimation of night-time minimum $T$, suggesting that land-surface processes may play a role. Of the 101 stations with $T$ measurements available, the dominant land-use categories of the grid cells containing stations are: *'Urban'* (10 sites); *'Dryland Cropland and Pasture'* (4 sites); *'Grassland'* (72 sites); *'Deciduous Broadleaf Forest'* (1 sites); *'Evergreen Needleleaf Forest'* (11 sites); and, *'Mixed Forest'* (3 sites). The overestimation of night-time $T$ is greatest at stations located in grid cells classified as urban (Fig. 6b), consistent with a previous evaluation of WRF with the Noah-MP LSM for urban and rural stations in summer (Salamanca et al., 2018). The bias amplification in urban grid cells may reflect an incorrect classification of the underlying land surface in WRF, as only the München-Stadt station (id 3379) is listed as an urban station on the DWD's list for computing heat island effects. It may also result from an overestimation of heat storage when a mosaic approach is not used, and therefore the entire grid cell is treated as urban (Daniel Fenner, personal communication). The potential role of the land-surface specification or properties is reinforced by the comparison with MODIS data (Fig. 7), which shows the largest warm biases over grid cells classified as urban or croplands while biases are smaller in forested areas. There is also a cold bias along the foothills and at higher elevations in the Alps. The biases are slightly smaller in WRF_NUDGE than in WRF_NO_NUDGE, consistent with the station-based assessment.

In addition to factors internal to WRF, we note that the driving reanalysis data may also contribute to the warm bias, at least at some locations. From the available observations, 60 stations have both valid $T$ data between June and August 2018 and a modelled elevation in ERA5 that is within ±100 m of reality. Averaged over the summer months and all stations, ERA5 has

a mean warm bias of 0.37°C. At 25 of the sites, a warm bias exceeding 0.5°C is present, with an average value over these sites of 0.92°C.

The inclusion of grid-analysis nudging leads to a small but nearly uniform improvement in agreement between observed and simulated variables. The distribution of MDs is closer to zero for all variables except *U* and *PS*, while those of MADs are
closer for all variables (cf. Fig. 3 and Table 3). $R^2$ values are also uniformly higher when nudging is used, and the lowest lower-quartile value is 0.3 in WRF_NUDGE compared with only 0.18 in WRF_NO_NUDGE. Nudging produces a particularly noticeable improvement in simulated precipitation, halving the MD and nearly doubling the $R^2$ values (cf. Fig. 3, Fig. 4 and Table 3). Its usage also reduces the magnitude of the seasonal temperature biases and the number of extreme occurrences of the warm bias in summer (cf. Fig. 4 and Fig. 6). Considering daily timescales, the agreement of
WRF_NUDGE with the observations is similar or even improved (Table 4): the mean MD is largely unaffected, but the average MAD decreases and average $R^2$ increases. Based on these improvements, grid-analysis nudging was adopted for the climatological simulations.

### 3.2 Evaluation of BAYWRF

Averaged over the full simulation period, BAYWRF shows a similar magnitude of agreement with station *T* and *RH* data at
daily timescales as found at sub-diurnal timescales for the single evaluation year (Fig. 8; cf. Fig. 3). For *T*, the MD has lower and upper quartiles of -0.3 and 0.4°C, respectively, while the values of $R^2$ uniformly exceed 0.92. For *RH,* the MD has lower and upper quartiles of 0.4 and 4.4% while the respective values for $R^2$ are 0.57 and 0.65. For *PREC*, the upper and lower quartiles of MDs considering days with observed precipitation are -0.1 and 0.1 mm while for $R^2$ the values are 0.41 and 0.47. A similar number and magnitude of wet false events are simulated (twenty times less than the sample size of observed
events). Spatially, BAYWRF exhibits a positive bias in *T* and a negative bias in *RH* in the interior of Bavaria, and the converse anomalies in the pre-alpine and alpine areas in the south and along the eastern border of the region (Fig. 8a, c). The mean $R^2$ values for *RH* show a clear meridional gradient (Fig. 8d), which suggests that the model has some difficulty capturing processes governing near-surface moisture fluctuations in the southern part of Bavaria. Nonetheless, the highest correlation coefficients for observed precipitation events are found in this region (Fig. 8f). In addition, considering monthly
precipitation sums, the centered pattern correlation between REGNIE and BAYWRF ranges from a lower quartile of 0.64 to an upper quartile of 0.82. Therefore, the characteristics of precipitation variability in time and space are captured by the dataset.

For BAYWRF, we note that in addition to the potential factors contributing to temperature biases discussed in Section 3.1,
evaluation of the climatological simulation is also affected by discontinuities in station location and instrumentation. One example is Nürnberg (id 3668), which moved on 4 December 1995 from (49.4947ºN, 11.0806ºE) to (49.5030ºN, 11.0549ºE). The older station position is shifted one grid cell to the south and one grid cell to the west compared with its current location,

corresponding to a shift in land use from urban (old position) to grasslands (new). Any discontinuities in location and underlying surface type are not captured since the most recent station positions are used for extracting meteorological data from D2. This potential source of discrepancies should be taken into consideration for climatological analyses (e.g., comparing observed and simulated trends).

## 4 Data Availability

Data from BAYWRF are available for download on the Open Science Framework (OSF; Collier, 2020; https://www.doi.org/10.17605/OSF.IO/AQ58B). Due to the size of the simulations, we have only provided daily mean data from the finest WRF domain (D2; 1.5-km grid spacing) after cropping close to the extent of Bavaria and removing vertical levels above ~ 200 hPa, amounting to 450 GB in total. Data are divided into three- and four-dimensional fields by year and month, with respective file sizes of ~150 MB and 1.1 GB. For the four-dimensional data, perturbation and base-state atmospheric pressure (WRF variables P and PB) and geopotential (PH and PHB) were combined to generate full model fields, while perturbation potential temperature (T) was converted to atmospheric temperature.

## 5 Conclusions

We presented a climatological kilometer-scale simulation with the atmospheric model WRF over Bavaria for the period of September 1987 to August 2018. Comparison of simulations for the period of September 2017 to August 2018 with and without grid-analysis nudging against extensive meteorological measurements across Bavaria showed that nudging decreased the mean deviations and increased the coefficient of determinations at the majority of sites for nearly all evaluated atmospheric variables, in particular precipitation. This approach was therefore adopted for generating the full BAYWRF dataset. In general, BAYWRF represents the variability of near-surface meteorological conditions well, albeit with both seasonal and spatial biases that are explored briefly here. Future users of this dataset are encouraged to rigorously evaluate biases for the variables and time periods relevant to their particular study areas and applications. BAYWRF provides a useful database for linking large-scale climate, as represented by the ERA5 reanalysis, to mesoscale climate over Germany, to local conditions in Bavaria, in a physically based way. The data are intended for dendroecological research applications but would also provide a valuable tool for investigations of the climate dependence of economic, societal, ecological, and agricultural processes in Bavaria.

## 6 Appendix A: Sample WRF namelist

&time_control
run_days                    = 31,

```
      run_hours              = 0,
      run_minutes            = 0,
      run_seconds            = 0,
      start_year             = 2018, 2018,
start_month            = 08,  08,
      start_day              = 01,  01,
      start_hour             = 00,  00,
      start_minute           = 00,  00,
      start_second           = 00,  00,
end_year               = 2018, 2018,
      end_month              = 09,  09,
      end_day                = 01,  01,
      end_hour               = 00,  00,
      end_minute             = 00,  00,
end_second             = 00,  00,
      interval_seconds       = 10800,
      input_from_file        = .true.,.true.,
      history_interval       = 120,  120,
      frames_per_outfile     = 12, 12,
restart                = .true.,
      restart_interval       = 44640,
      override_restart_timers = .true.,
      write_hist_at_0h_rst   = .true.,
      io_form_history        = 2
io_form_restart        = 102
      io_form_input          = 2
      io_form_boundary       = 2
      debug_level            = 0
      auxinput4_inname       = "wrflowinp_d<domain>",
auxinput4_interval     = 180,
      io_form_auxinput4      = 2
      /
      &domains
      time_step              = 45,
```

```
time_step_fract_num      = 0,
      time_step_fract_den      = 1,
      max_dom                  = 2,
      e_we                     = 351, 351,
      e_sn                     = 301, 351,
e_vert                   = 60,  60,
      auto_levels_opt          = 2,
      max_dz                   = 600.,
      dzbot                    = 20.,
      dzstretch_s              = 1.5,
dzstretch_u              = 1.3,
      p_top_requested          = 1000,
      num_metgrid_levels       = 33,
      num_metgrid_soil_levels  = 4,
      dx                       = 7500, 1500,
dy                       = 7500, 1500,
      grid_id                  = 1,    2,
      parent_id                = 0,    1,
      i_parent_start           = 1,    145,
      j_parent_start           = 1,    126,
parent_grid_ratio        = 1,    5,
      parent_time_step_ratio   = 1,    5,
      feedback                 = 0,
      smooth_option            = 0,
      /
&physics
      mp_physics               = 10,   10,
      ra_lw_physics            = 4,    4,
      ra_sw_physics            = 4,    4,
      radt                     = 5,    5,
sf_sfclay_physics        = 1,    1,
      sf_surface_physics       = 4,    4,
      bl_pbl_physics           = 1,    1,
      topo_wind                = 1,    1,
```

```
        bldt                    = 0,    0,
cu_physics              = 1,    0,
        cudt                    = 0,    0,
        ysu_topdown_pblmix      = 1,
        isfflx                  = 1,
        ifsnow                  = 1,
num_soil_layers         = 4,
        num_land_cat            = 24,
        sf_urban_physics        = 0,    0,
        slope_rad               = 0,    1,
        topo_shading            = 0,    1,
cu_rad_feedback         = .true.,.true.,
        usemonalb               = .true.,
        bucket_mm               = 100.,
        sst_update              = 1,
        tmn_update              = 1,
lagday                  = 150,
        sst_skin                = 1,
        /
        &noah_mp
        opt_alb                 = 2,
opt_snf                 = 1,
        dveg                    = 5,
        /
        &fdda
        grid_fdda               = 1,0,
gfdda_inname            = "wrffdda_d<domain>",
        gfdda_interval_m        = 180, 180,
        gfdda_end_h             = 100000, 100000,
        io_form_gfdda           = 2,
        fgdt                    = 0, 0, 0,
if_no_pbl_nudging_uv    = 1, 0, 0,
        if_no_pbl_nudging_t     = 1, 0, 0,
        if_no_pbl_nudging_q     = 1, 0, 0,
```

```
      if_zfac_uv          = 1,  0,  0,
      k_zfac_uv           = 10, 0,  0,
if_zfac_t           = 1,  0,  0,
      k_zfac_t            = 10, 0,  0,
      if_zfac_q           = 1,  0,  0,
      k_zfac_q            = 10, 0,  0,
      guv                 = 0.0003, 0.0, 0.0,
gt                  = 0.0003, 0.0, 0.0,
      gq                  = 0.00005, 0.0, 0.0,
      if_ramping          = 0,
      /
      &dynamics
w_damping           = 0,
      diff_opt            = 2,    2,
      km_opt              = 4,    4,
      diff_6th_opt        = 0,    0,
      diff_6th_factor     = 0.12,  0.12,
base_temp           = 290.
      damp_opt            = 3,
      zdamp               = 5000.,  5000.,
      dampcoef            = 0.2,   0.2,
      khdif               = 0,    0,
kvdif               = 0,    0,
      non_hydrostatic     = .true., .true.,
      moist_adv_opt       = 2,    2,
      scalar_adv_opt      = 2,    2,
      epssm               = 0.2,   0.5,
mix_full_fields     = .true.,
      /
      &bdy_control
      spec_bdy_width      = 5,
      spec_zone           = 1,
relax_zone          = 4,
      specified           = .true., .false.,
```

```
nested                  = .false., .true.,
/
&namelist_quilt
nio_tasks_per_group     = 0,
nio_groups              = 1,
/
```

## 7 Author contributions

EC performed the simulations, analyzed the data and wrote the manuscript. TM developed the study concept and wrote the
manuscript.

## 8 Competing interests

The authors declare that they have no conflict of interest.

## 9 Acknowledgements

This project is sponsored by the Bavarian State Ministry of Science and the Arts in the context of the Bavarian Climate
Research Network (bayklif). We gratefully acknowledge the compute resources and support provided by the Erlangen
Regional Computing Center (RRZE) and we thank Thomas Zeiser for his assistance with the timely completion of the
simulations.

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

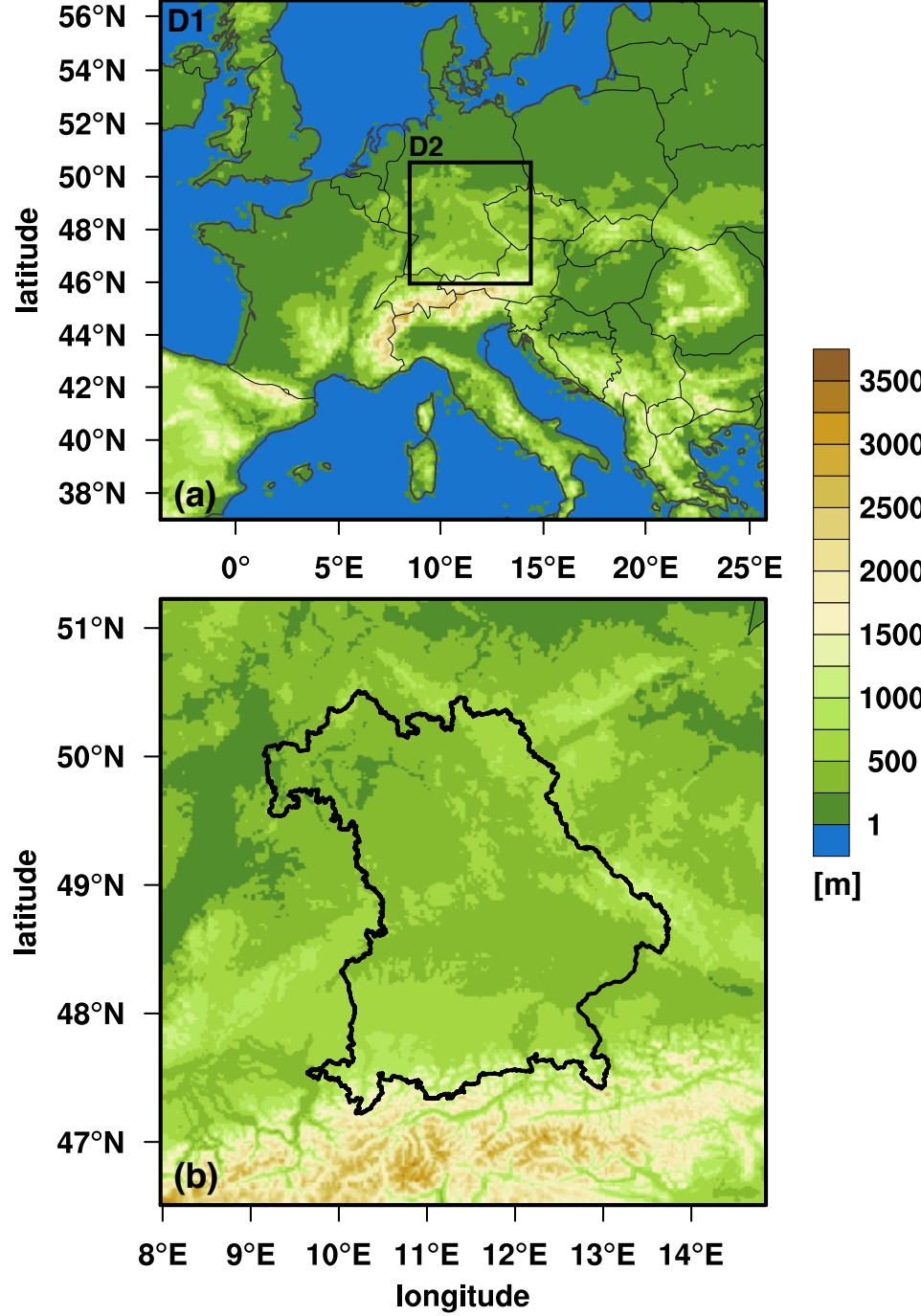


**Figure 1:** Extent and modelled topographic height in WRF D1 (a) and D2 (b). The extent of D2 and of Bavaria are delineated in black in the top and bottom panels, respectively.

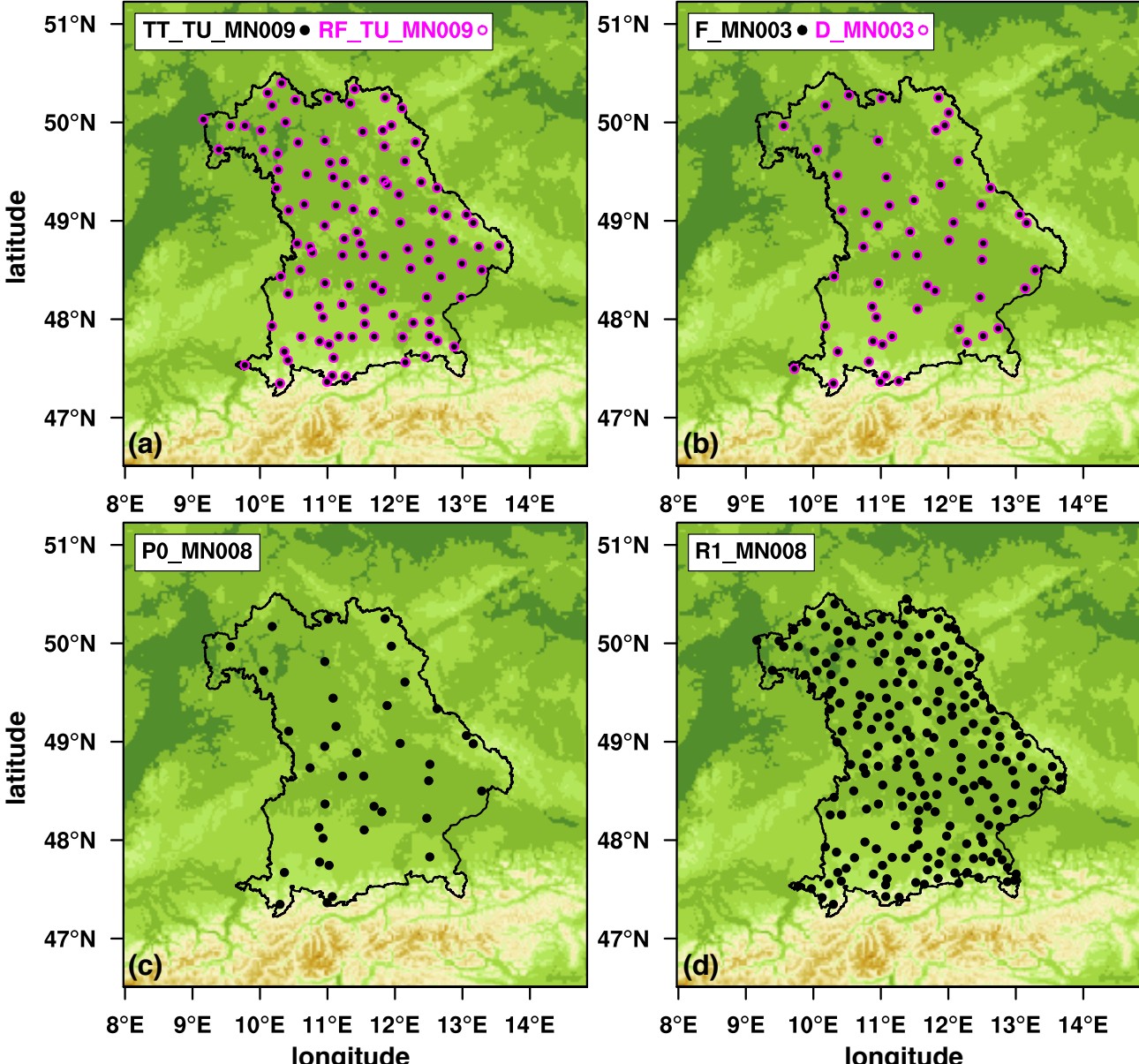

**Figure 2:** The location of the stations used for model evaluation during the most recent simulation year (September 2017 to August 2018) for each dataset listed in Table 2. Datasets labelled in black are shown by filled black circles, while datasets labelled in pink are shown by open pink circles, illustrating that locations for measurements of air temperature and humidity (a; TT_TU_MN009 & RF_TU_MN009) and of wind speed and direction (b; F_MN003 & D_MN003) were the same. The locations for measurements of surface pressure (P0_MN008) and of precipitation (R1_MN008) are shown in panels c and d, respectively.

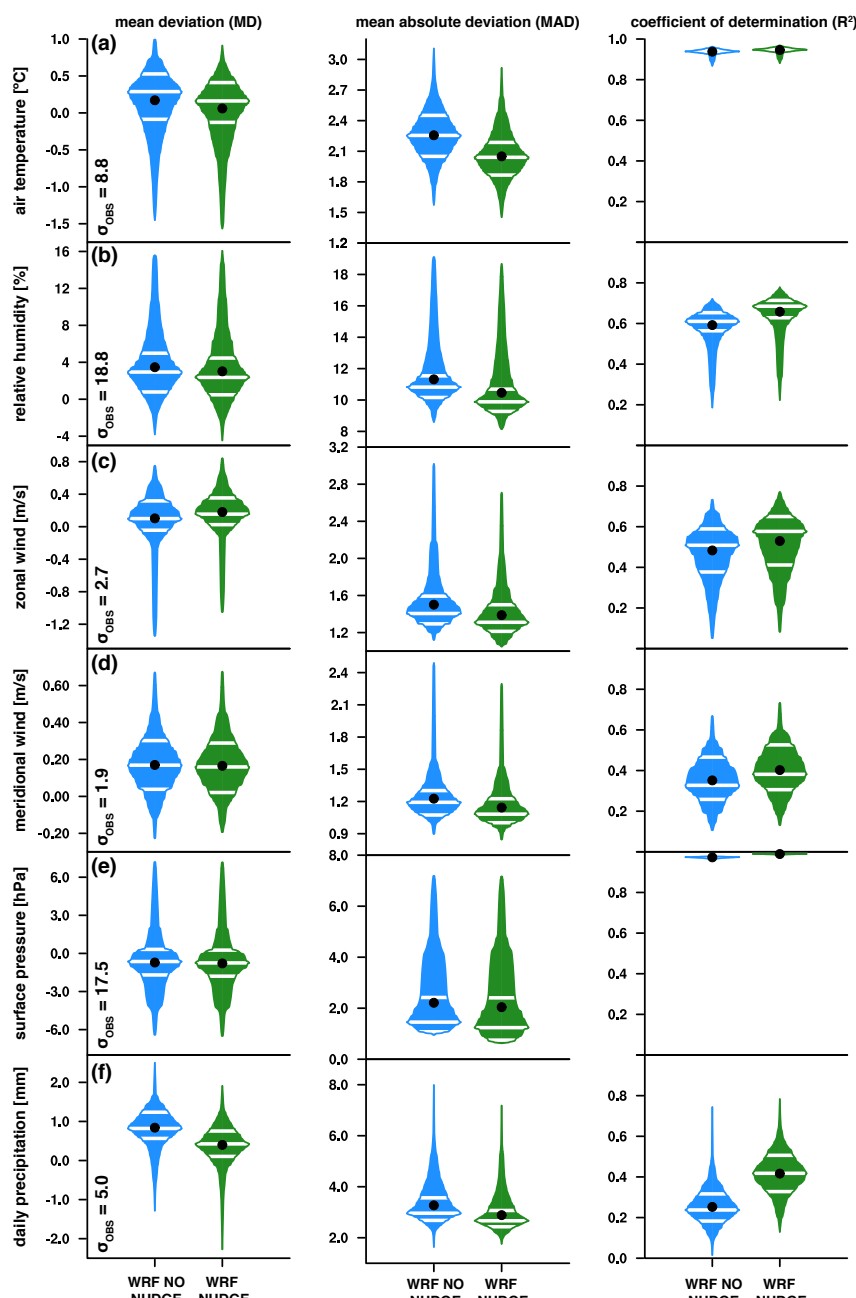

**Figure 3:** Box-percentile plots (Esty and Banfield, 2003) of mean deviation (MD), mean absolute deviation (MAD), and coefficient of determination ($R^2$) between observations and the two WRF simulations, WRF_NO_NUDGE (blue) and WRF_NUDGE (green), for 2-m (a) air temperature and (b) relative humidity, 10-m (c) zonal and (d) meridional winds, (e) surface pressure and (f) precipitation. The statistics for all variables except for precipitation were computed from two-hourly instantaneous values, while those for precipitation were computed using daily totals. The shape of the plots shows the distribution of data over their range of values, white lines delineate 25th, 50th and 75th percentiles, and a black dot indicates the mean. The observed standard deviation ($\sigma_{obs}$) for each variable is provided in the left column.

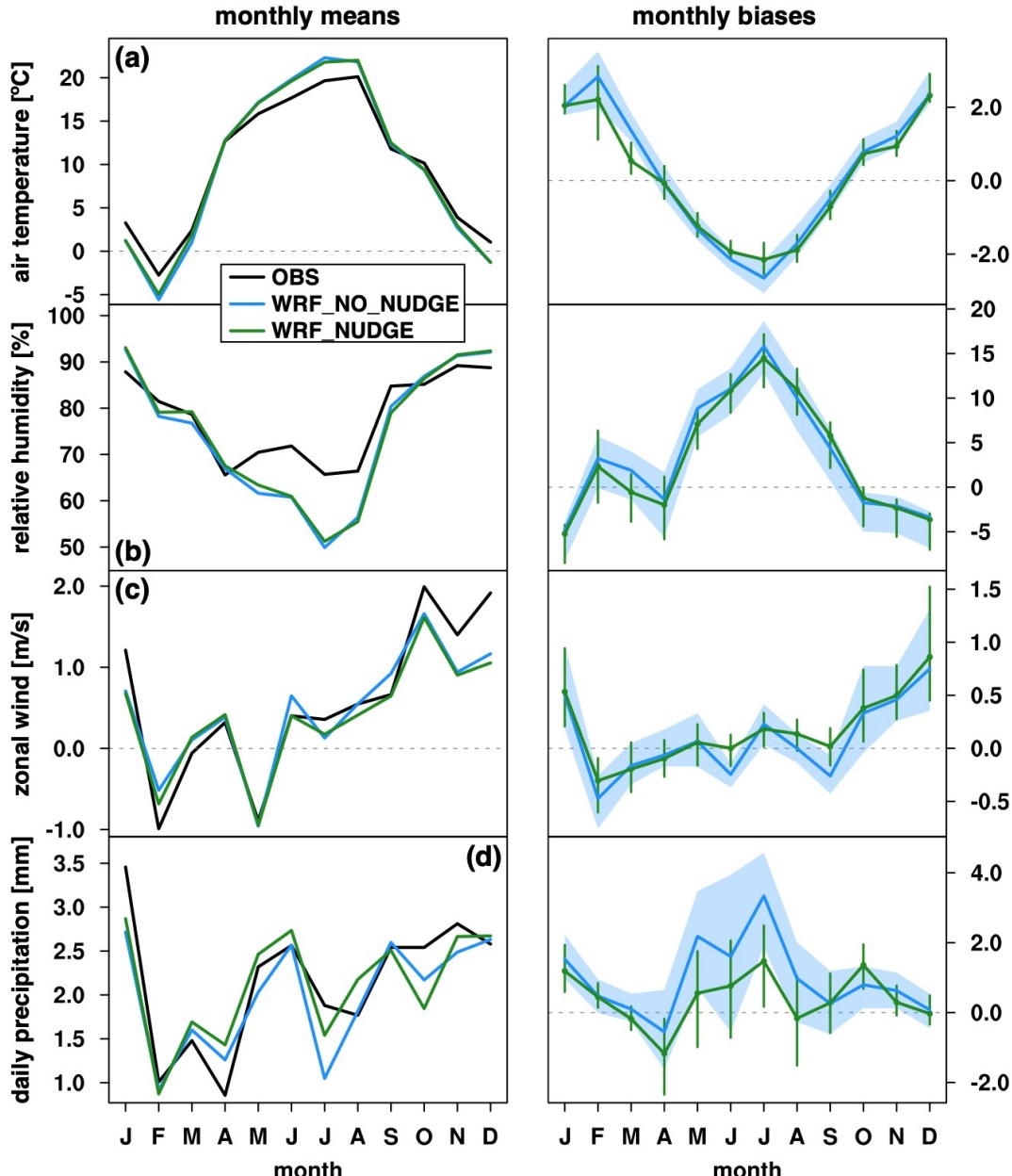


**Figure 4:** Timeseries of monthly mean 2-m (a) air temperature and (b) relative humidity, (c) 10-m zonal winds, and (d) daily total precipitation (left column) between September 2017 and August 2018. Observational, WRF_NO_NUDGE and WRF_NUDGE data are shown in black, blue and green, respectively. Timeseries of monthly mean biases of the same variables (right column). The mean bias over all stations is shown for each simulation using the same colour assignment, while the lower and upper quartile of the station biases is shown as a blue polygon and green bars for WRF_NO_NUDGE and WRF_NUDGE data, respectively.

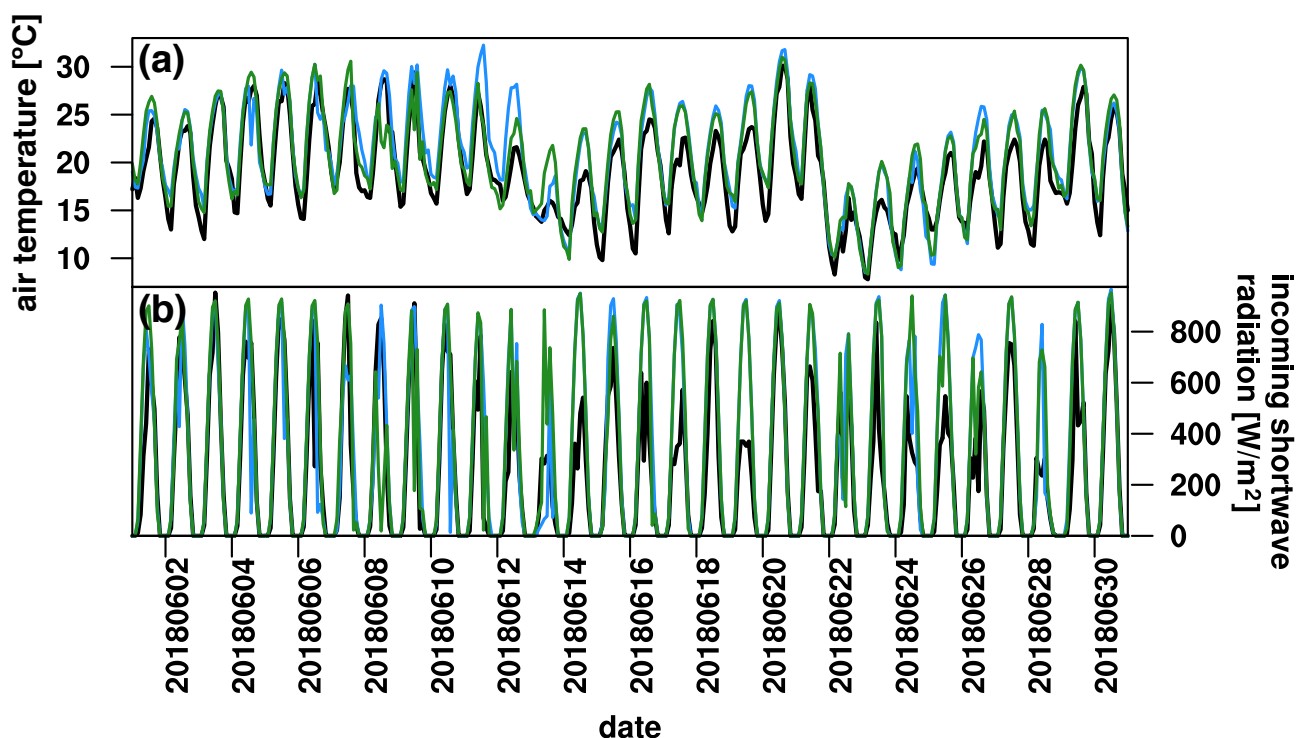

**Figure 5:** Timeseries of (a) 2-m air temperature and (b) incoming shortwave radiation at the station in Nürnberg (id 3668) from 1 June to 1 July 2018. Observational, WRF_NO_NUDGE and WRF_NUDGE data are shown in black, blue and green, respectively.

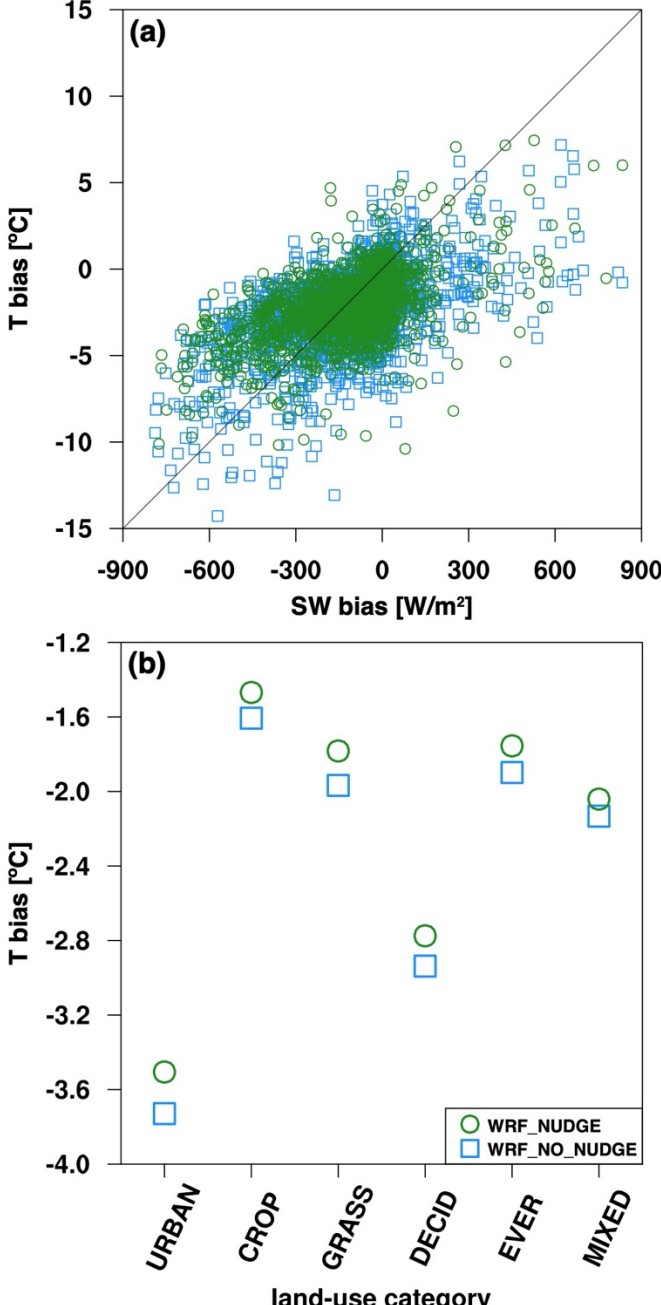

**Figure 6:** Scatter plots of (a) air temperature bias vs. incoming shortwave radiation bias and (b) air temperature bias vs. land-use category in closest grid cell to station. The category abbreviations from left to right describe: 'Urban and Built-Up Land' (10 sites); 'Dryland Cropland and Pasture' (4 sites); 'Grassland' (72 sites); 'Deciduous Broadleaf Forest' (1 sites); 'Evergreen Needleleaf Forest' (11 sites); and, 'Mixed Forest' (3 sites). For both panels, data from WRF_NO_NUDGE and WRF_NUDGE are displayed as blue square and green circle markers, respectively.

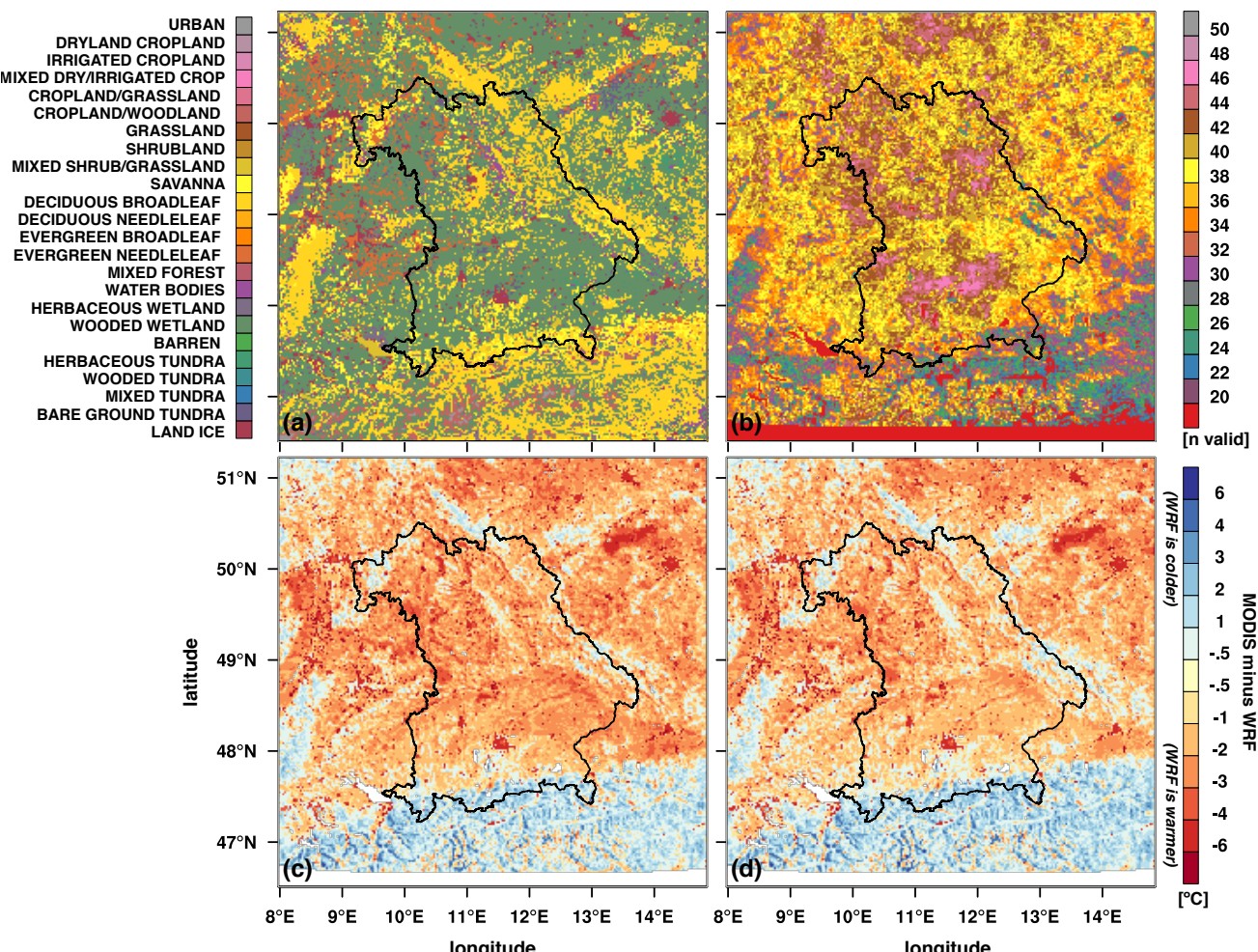

**Figure 7:** (a) Land-use classification in D2. (b) Number of timesteps with valid night-time LST data in the MODIS MYD11A1 dataset between 1 June and 31 August 2018 out of a maximum of 52 with less than 50% missing data in D2. The average difference in observed and simulated LST for (c) WRF_NO_NUDGE and (d) WRF_NUDGE. Note that the orange and red colours in panels c and d shade areas where WRF is warmer than MODIS (MODIS minus WRF is negative) and vice versa for blues.

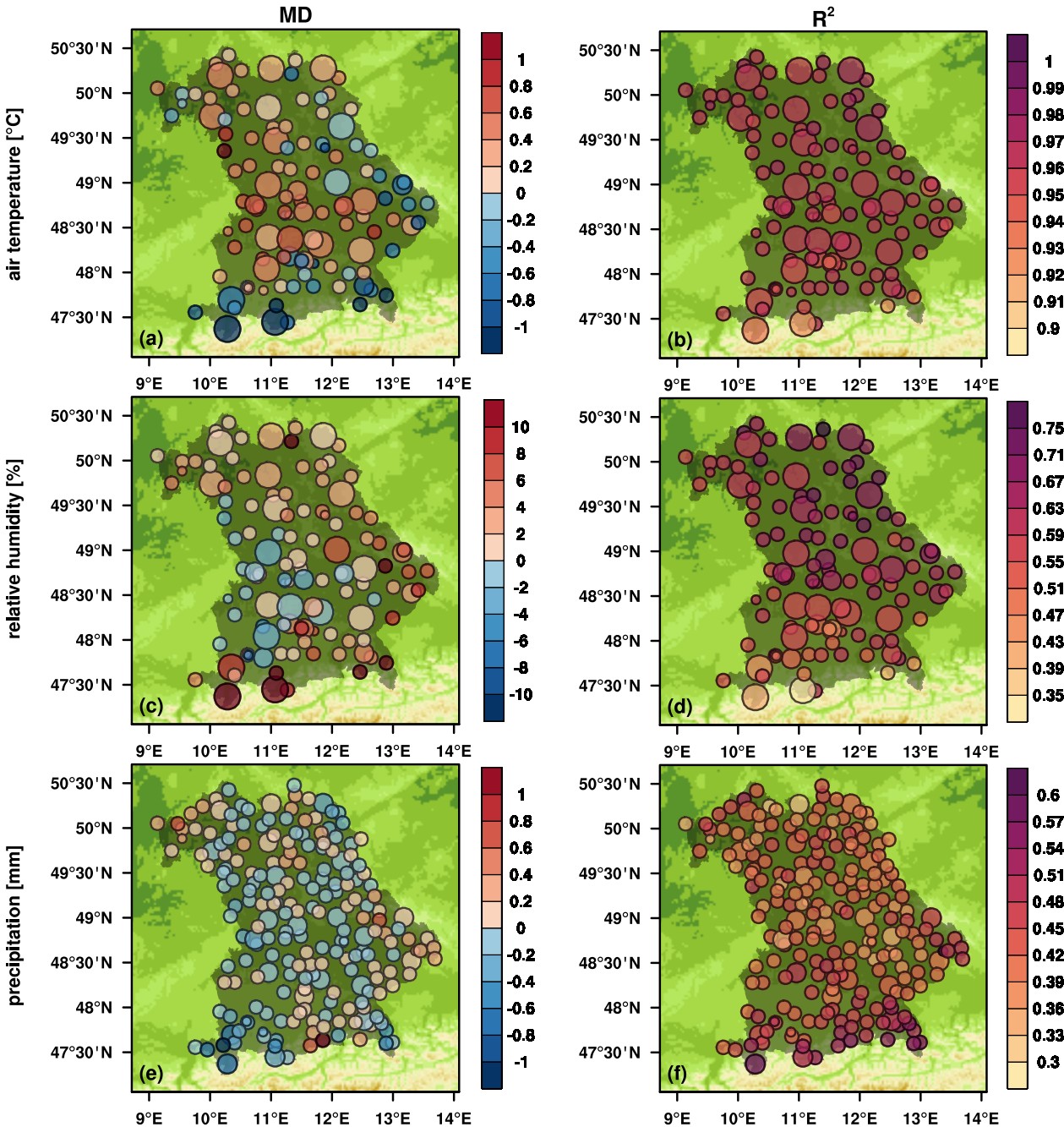

**Figure 8:** Spatial maps of mean MD (left column) and $R^2$ (right) at all stations with valid data between September 1987 and August 2018 for daily mean (a, b) *T* and (c, d) *RH,* and for daily total (e, f) *PREC*. The four marker sizes group the percentage of the total timesteps (11,323 days) for which data were available at each station into the four quartiles. The largest marker size, which delineates records with more than 75% valid data points, is therefore not available for *PREC*, as this dataset begins on 1 September 1995.

**Table 1: Summary of the WRF configuration used for BAYWRF.** A full sample namelist is provided in Appendix A.

| Table 1: WRF configuration | | |
|---|---|---|
| *Domain configuration* | | |
| Horizontal grid spacing | 7.5 & 1.5 km (D1–2) | |
| Grid dimensions | 351x301, 351x351 | |
| Time step | 45 & 9 s | |
| Vertical levels | 60 | |
| Model top pressure | 10 hPa | |
| *Model physics* | | |
| Radiation | RRTMG | (Iacono et al., 2008) |
| Microphysics | Morrison | (Morrison et al., 2009) |
| Cumulus | Kain-Fritsch (none in D2) | (Kain, 2004) |
| Planetary boundary layer | Yonsei State University | (Hong et al., 2006) |
| Atmospheric surface layer | Monin Obukhov | (Jiménez et al., 2012) |
| Land surface | Noah-MP | (Niu et al., 2011) |
| *Dynamics* | | |
| Top boundary condition | Rayleigh damping | |
| Diffusion | Calculated in physical space | |

590 **Table 2: A summary of data used for model evaluation.** Rows highlighted in grey provide information about observational data from the DWD CDC Data Portal, whose measurement locations for the evaluation for the 2017 to 2018 period are shown in Figure 2.

| Dataset Name | Variable [unit] | Temporal Resolution | Total Stations in Bavaria 2017-2018 (1987-2018) | Version | Access URL | Last Accessed | Dataset Description |
|---|---|---|---|---|---|---|---|
| TT_TU_MN009 | 2-m air temperature [°C] | Hourly | 106 (120) | v19.3 | https://cdc.dwd.de/portal | | https://cdc.dwd.de/sdi/pid/TT_TU_MN009/DESCRIPTION_TT_TU_MN009_en.pdf |
| RF_TU_MN009 | 2-m relative humidity [%] | Hourly | 106 (120) | v19.3 | https://cdc.dwd.de/portal | | https://cdc.dwd.de/sdi/pid/RF_TU_MN009/DESCRIPTION_RF_TU_MN009_en.pdf |
| F_MN003 | 10-m wind speed [m/s] | Hourly | 57 | v19.3 | https://cdc.dwd.de/portal | | https://cdc.dwd.de/sdi/pid/F_MN003/DESCRIPTION_F_MN003_en.pdf |
| D_MN003 | 10-m wind direction [deg] | Hourly | 57 | v19.3 | https://cdc.dwd.de/portal | | https://cdc.dwd.de/sdi/pid/D_MN003/DESCRIPTION_D_MN003_en.pdf |
| P0_MN008 | surface pressure [hPa] | Hourly | 38 | v19.3 | https://cdc.dwd.de/portal | 10 Sep 2020 | https://cdc.dwd.de/sdi/pid/P0_MN008/DESCRIPTION_P0_MN008_en.pdf |
| R1_MN008 | precipitation [mm] | Hourly | 213 (219) | v19.3 | https://cdc.dwd.de/portal | | https://cdc.dwd.de/sdi/pid/R1_MN008/DESCRIPTION_R1_MN008_en.pdf |
| Hourly station observations of solar incoming (total/diffuse) and longwave downward radiation for Germany | Incoming longwave and shortwave radiation [J/cm2] | Hourly | 10 | recent | https://cdc.dwd.de/portal | | https://opendata.dwd.de/climate_environment/CDC/observations_germany/climate/hourly/solar//DESCRIPTION_obsgermany_climate_hourly_solar_en.pdf |
| REGNIE | precipitation [mm] | Monthly sum | -- | recent | https://opendata.dwd.de/climate_environment/CDC/grids_germany/monthly/regnie/ | | https://opendata.dwd.de/climate_environment/CDC/grids_germany/monthly/regnie/DESCRIPTION_gridsgermany_monthly_regnie_en.pdf |
| MODIS MYD11A1 | land surface temperature [K] | Daily | -- | v006 | https://lpdaacsvc.cr.usgs.gov/appeears | | https://lpdaac.usgs.gov/products/myd11a1v006/ |

595

**Table 3: A summary of the statistical evaluation of the WRF_NO_NUDGE (italics) and WRF_NUDGE (bold italics) simulations, considering the whole evaluation period of 1 September 2017 to 1 September 2018. The table presents the mean deviation (MD), the mean absolute deviation (MAD) and the coefficient of determination ($R^2$) for two-hourly 2-m air temperature (*T*) and relative humidity *RH*), 10-m zonal wind (*U*) and meridional wind (*V*), surface pressure (*PS*), and daily total precipitation (*PR*). All computations are made from observations minus model data.**

| Variable | MD | MAD | R2 |
|---|---|---|---|
| *T (WRF_NO_NUDGE)* | 0.2 | 2.3 | 0.94 |
| ***T (WRF_NUDGE)*** | **0.1** | **2.0** | **0.95** |
| *RH* | 3.5 | 11.3 | 0.59 |
| ***RH*** | **3.0** | **10.5** | **0.66** |
| *U* | 0.1 | 1.5 | 0.48 |
| ***U*** | **0.2** | **1.4** | **0.53** |
| *V* | 0.2 | 1.2 | 0.35 |
| ***V*** | **0.2** | **1.1** | **0.40** |
| *PS* | -0.7 | 2.2 | 0.97 |
| ***PS*** | **-0.8** | **2.0** | **0.99** |
| *PR* | 0.8 | 3.3 | 0.25 |
| ***PR*** | **0.4** | **2.9** | **0.42** |

**Table 4: Same as Table 3 but for daily mean variables in WRF_NUDGE only.**

| Variable | MD | MAD | R2 |
|---|---|---|---|
| ***T*** | **0.1** | **1.7** | **0.97** |
| ***RH*** | **3.0** | **8.4** | **0.71** |
| ***U*** | **0.2** | **0.9** | **0.72** |
| ***V*** | **0.2** | **0.6** | **0.64** |
| ***PS*** | **-0.8** | **2.0** | **0.99** |