# Peer review of "BAYWRF: a high-resolution present-day climatological atmospheric dataset for Bavaria"

_Earth System Science Data, 2020_

## Referee Comment (RC1) · Benjamin Poschlod (Referee) · 8 Jul 2020

General comments

The manuscript by Emily Collier & Thomas Mölg gives a comprehensive overview of a high-resolution 30-year climatological data set over Bavaria. The climate simulations were produced by the WRF model in 1.5 km resolution, nested in a 7.5-km-resolution domain and driven by ERA5 boundary conditions. The authors evaluate the model performance for air temperature, relative humidity, winds, surface pressure, precipitation, and land surface temperature for a 12-month period where they compare simulated values to observational data. Additionally, the effect of the application of nudging is

assessed. Generally, the manuscript is well-written, and the figures support the presentation of the data set and its evaluation. In particular, the authors' handling of errors in the data set (e.g. sub-surface temperature in single glacier pixels) and explanation of deviations/biases (e.g. urban heat islands, connection between overestimated air temperature and overestimated radiation) are very valuable features of the data description. The data are easily accessible and valuable for further application with focus on impact-related studies. Though, the total size of the 30-year daily-resolution data set (~ 450 GB) may not be easy to handle for users, who are new to the application of high-resolution climate data. On the other hand, users from the field of climate science would be interested in even higher temporal resolution, especially regarding the precipitation data. In sum, I consider the manuscript and the data appropriate for the publication within ESSD, although I recommend minor revisions based on the following remarks.

Specific comments

L1: Title: the data set is described as "convection-resolving". Though, within the whole manuscript, no convective events have been evaluated. Furthermore, the data set is provided in daily resolution, which is why short convective events cannot be investigated properly. Hence, I would suggest replacing "convection-resolving" by "high-resolution".

L80 / Table 1: The Kain-Fritsch cumulus scheme is applied for the 7.5 km domain, but not for the 1.5 km domain. According to that, not only deep, but also shallow convection is explicitly resolved in the 1.5 km domain? I would suggest clarifying this in the text.

L143: "For the distributed trend analysis, we did not apply a field significance test (e.g., Wilks, 2016) due to the small sample size." – Does the "distributed trend analysis" refer to the results in L241 – 246 and Fig. 9? If yes – can you explain why is the sample size too small? If you test the trend at all 351x351 locations, the p-value should be adjusted for statistical tests at many locations (following e.g. Wilks 2016). Moreover,

the reference (Wilks, 2016) is missing in your Reference section. Please also clarify, which test or method you used to detect trends.

L180: Has the observational precipitation data from DWD been corrected for under-catch? Especially in (pre-)alpine regions, this plays a major role, in particular for solid precipitation. I would recommend to briefly discuss this source of uncertainty.

L385: Figure 4 gives a good overview of the biases averaged for all locations. Though, the spatial distribution of biases would be of high interest as well. As the manuscript is already quite long and contains many figures, I would suggest creating such bias maps and moving these additional figures to a supplementary file.

Technical corrections

L156: 273.16 unit is missing

Figures 3,4,5,7,8: Temperature unit is "C" instead of "$^\circ$C". Figure 6: Here the unit is missing in the figure (and given in the caption instead) Figure 9: Here you use "K" –> Please unify

---

## Referee Comment (RC2) · Michael Warscher (Referee) · 9 Jul 2020

In their manuscript "BAYWRF: a convection-resolving, present-day climatological atmospheric dataset for Bavaria", the authors present a new high-resolution RCM simulation using WRF and ERA5 reanalysis data as boundary condition. They evaluate the performance for the target region of Bavaria using station observations.

General Comments

The manuscript is very well written and of high technical and scientific quality. It fits very well in the scope of ESSD. However, I have several issues, questions, and suggestions

which at large could lead to major revisions. However, I understand that the manuscript is mainly an overview of the presented data and thereby, the amount and detail of the analyses and following content has to be limited at certain points. I would gladly leave the decision to the editors on how much of and at what detail level my suggestions should be addressed. The dataset they produced is generally very valuable for the scientific community, as well as for many users in different sectors.

The authors have chosen a single year as specific validation period. In addition, they point out in L. 101 that it is not an average year in terms of seasonal climatology (record heatwave in 2018). The chosen year might therefore not be a representative period for the RCM performance in other years. However, an extension of the evaluation period seems just limited by a missing run using the NO_NUDGE configuration. As the whole exercise is a historic / present-day reanalysis driven simulation effort, the NUDGE setup was run for the whole 30-year period anyways and would be available for additional validation years. I would highly recommend to add at least one additional year of validation (the more the better) to strengthen the results under different conditions. The validation could potentially even be done for the whole 30-years (just being limited by available observations and – by now - the missing NO_NUDGE for more than one year). I am also quite sure that an extension of the analysis would not limited by available observation data. If no additional simulations (NO_NUDGE) can be performed, the authors might think of some additional validation using the 30-year NUDGE run only.

While I really acknowledge the direct comparison to station data, the study would highly benefit from a comparison to gridded observation data sets such as REG-NIE (1 km, https://www.dwd.de/DE/leistungen/regnie/regnie.html) or HYRAS (5 km, https://www.dwd.de/DE/leistungen/hyras/hyras.html). Besides the correct representation of single stations, the real benefit of such a computationally expensive high-resolution simulation might or should be – besides the reproduction of observed station data - the resolving of spatial distributions.

The authors point out, that they use a convection-permitting resolution of 1.5 km. However, the topic of simulating convective precipitation is not referred to again in the manuscript. This is still a very important and relevant topic, and the presented data would be ideally suited to look into this. Several questions arise and could easily be tackled. E.g. the authors show an underestimation of precipitation at some point. Could this be explained by the 1.5 km still being too coarse to resolve all or enough convective events? Are the results of the KF parameterization in the 7.5 km (D1) similar or totally different? Are sub-daily precipitation dynamics captured? Some of these questions could quite easily be investigated by comparing your results of D1 (convection parameterized) and D2 (convection resolved) to gridded precipitation products such as REGNIE and maybe even to station data.

This leads to another question regarding the resolution. While I do not question the validity and satisfying performance of the presented simulation, I would be very glad to see more about the added value of such a high resolution. This could be done by a comparison of the performance between the results of Domain D1 and Domain D2. There are no analyses in the manuscript that try to address this important question.

Another important issue regarding the trend analysis can be found in the specific comments.

Specific Comments

L. 12: I suggest to remove the reference to the project here (and at other positions in the manuscript) and state the project name solely in the acknowledgement section.

L. 32 – 38: I see that the linkage to dendroclimatological studies refers to the research project, but in my opinion, this is not needed here. You don't show any further results regarding this topic, and the general effort and method of dynamical downscaling does not really need to be justified or explained within this manuscript.

L. 49 – 59: The same as the comments above: this is in general interesting information,

but not within this manuscript. The paragraph should be shortened, maybe only keep the last sentence: "High-temporal..."

L. 66 – 68: Please remove the sentence: "These data..." for the reason stated above.

L. 69: While your statement here is certainly true, I would prefer a more moderate phrasing, e.g., "These data has the potential to find..." instead of "These data will also find...".

L. 78: Could you please give some more information on how the WRF configuration was chosen? This should then also be added to the manuscript. You state that the setup is based on Collier et al. (2019) but the study seems to be located in completely different climate and terrain conditions (East Africa). It is widely shown in literature that the performance of the chosen configuration strongly depends on the region. I understand that it is not feasible to perform a full configuration optimization ensemble, but some more information on this issue should be added.

L. 122: It would be very interesting to see sub-daily results also for precipitation from such a simulation. By permitting convective events, this could potentially be one of the strong points of such a high-resolution simulation.

L. 123: What about all the cases where modeled precipitation > 0 but the observed precipitation = 0? These cases should somehow be analyzed too and not be neglected in the performance analysis.

L. 128: Why did you choose two hourly WRF output? I see that the output somehow has to be confined, but hourly values would also be very valuable! Do you still have these available? I think it is fine for the manuscript to keep two hourly results, but at least for the main surface variables, it would be very useful to have hourly values as well. If they are available upon request, you could add this information to the data availability section.

L. 147 - 157: I highly appreciate the very well investigated and documented error handling here!

L. 160: See comment above: why did you choose two-hourly output? Was it just to save storage or is there another reason?

L. 169 - 170: It is very valuable that you try to compare the results to other studies (here to the work by Warscher et al. 2019), but the numbers are not really comparable here (different investigation area, nudging strategy, stations, terrain, etc.). I would either keep your statement and add an explanation regarding the differences in the analyses or remove the statement or phrase it differently ("similar but lower" is quite inexplicit).

L. 180 - 181: To me, this is a strong hint that the used resolution is still not high enough to correctly simulate absolute convective precipitation amounts. That's one reason why it would be so valuable to analyze more than one year of data and to compare results between D1 and D2.

L. 213: You clearly show that the grid-nudged run is performing better than the "free" simulation, which again leads to the question of the benefits of the simulation. This result indicates that WRF adds biases compared to the ERA5 forcing simulation when not grid-nudged to them. The DWD stations you used for your validation might even have been assimilated in ERA5 which again questions to some point the added value of the simulation.

L. 215 - 216 Remove "(the temporal resolution of data available in BAYWRF)". This is not important here.

L. 228: Three typos: add spaces after "WRF:"

Sect. 3.5: Trend analysis: it is quite obvious that the trends are reproduced by the simulation when it is forced by a reanalysis product such as ERA5 (and grid-nudging is used). You could think about removing the whole section, as I do not see a value in this information. If the trends would not have been reproduced, it would be an argument that something goes wrong, but – the other way round - these results are not

proving a good performance of WRF (as stated in the paragraph). You simply see the overall dynamics of the forcing (which includes assimilations of historic observations and therefore reproduces historic trends).

L. 241 - 246: The paragraph falls a bit short compared to the other ones. The spatial distribution of trends could be more elaborated (if a trend section is kept). The fine scale spatial differences of trends is in the end the information that is produced by the RCM simulations (see the statements regarding trends and reanalysis above).

L. 261 - 263: The statement regarding grid-nudging may be true, but I do not see it as a success, as the forcing obviously includes assimilated observations (see comments above).

Fig. 7 c) and d): If I understand it right, the values in the legend should be reversed (wrong sign).

---

## Author Comment (AC1) · 16 Sep 2020

Dear Benjamin Poschlod,

Thank you for your review of our manuscript and suggestions for improvement. Please find our replies to your comments below in blue.

Best regards,
Emily Collier & Thomas Mölg

**General comments**
The manuscript by Emily Collier & Thomas Mölg gives a comprehensive overview of a high-resolution 30-year climatological data set over Bavaria. The climate simulations were produced by the WRF model in 1.5 km resolution, nested in a 7.5-km-resolution domain and driven by ERA5 boundary conditions. The authors evaluate the model performance for air temperature, relative humidity, winds, surface pressure, precipitation, and land surface temperature for a 12-month period where they compare simulated values to observational data. Additionally, the effect of the application of nudging is assessed. Generally, the manuscript is well-written, and the figures support the presentation of the data set and its evaluation. In particular, the authors' handling of errors in the data set (e.g. sub-surface temperature in single glacier pixels) and explanation of deviations/biases (e.g. urban heat islands, connection between overestimated air temperature and overestimated radiation) are very valuable features of the data description.

The data are easily accessible and valuable for further application with focus on impact-related studies. Though, the total size of the 30-year daily-resolution data set (~450 GB) may not be easy to handle for users, who are new to the application of high-resolution climate data. On the other hand, users from the field of climate science would be interested in even higher temporal resolution, especially regarding the precipitation data. In sum, I consider the manuscript and the data appropriate for the publication within ESSD, although I recommend minor revisions based on the following remarks.
Thank you for your favorable assessment of our manuscript. Although the total dataset size is ~450 GB, the 3D variables that are most likely to be used for impacts assessments (e.g., near-surface air temperature, humidity and precipitation) amount to a more manageable 57 GB. With regards to the provision of higher temporal resolution precipitation data, please see our response to Michael Warscher for more details.

**Specific comments**
L1: Title: the data set is described as "convection-resolving". Though, within the whole manuscript, no convective events have been evaluated. Furthermore, the data set is provided in daily resolution, which is why short convective events cannot be investigated properly. Hence, I would suggest replacing "convection-resolving" by "high resolution".
We used the term "convection resolving" to describe the dataset following convention for atmospheric simulations with grid spacings below ~4-km. We did not mean to imply that we analyze convective events, however we agree that the use of this term could be misleading, especially to a wider audience, and therefore changed the title as suggested.

L80 / Table 1: The Kain-Fritsch cumulus scheme is applied for the 7.5 km domain, but not for the 1.5 km domain. According to that, not only deep, but also shallow convection is explicitly resolved in the 1.5 km domain? I would suggest clarifying this in the text.
Yes, as no additional parameterization is employed, deep and shallow convection are explicitly represented in the 1.5 km domain. We added to Sect. 2.1 "As no cumulus

parameterization was employed in D2, both deep and shallow convection are assumed to be explicitly resolved.

L143: "For the distributed trend analysis, we did not apply a field significance test (e.g., Wilks, 2016) due to the small sample size." – Does the "distributed trend analysis" refer to the results in L241 – 246 and Fig. 9? If yes – can you explain why is the sample size too small? If you test the trend at all 351x351 locations, the p-value should be adjusted for statistical tests at many locations (following e.g. Wilks 2016). Moreover, the reference (Wilks, 2016) is missing in your Reference section. Please also clarify, which test or method you used to detect trends.
Here we were referring to the sample size of years. Please note that based on the suggestion of the other reviewer, we removed the trends analysis from the manuscript in favor of expanding the model evaluation. Please see our response to this reviewer for more details.

L180: Has the observational precipitation data from DWD been corrected for undercatch? Especially in (pre-)alpine regions, this plays a major role, in particular for solid precipitation. I would recommend to briefly discuss this source of uncertainty.
We did not correct precipitation for undercatch and have added this information in Section 2.2. We also added to Section 3.1: "The MD is positive at the majority of stations, indicating that WRF generally underestimates observed precipitation. The underestimate is likely greater than reported here, since the observations were not corrected for wind-induced undercatch."

L385: Figure 4 gives a good overview of the biases averaged for all locations. Though, the spatial distribution of biases would be of high interest as well. As the manuscript is already quite long and contains many figures, I would suggest creating such bias maps and moving these additional figures to a supplementary file.
We added some spatially distributed bias analysis as part of the expanded model evaluation.

**Technical corrections**
L156: 273.16 unit is missing
We changed to "exceeded the melting point."

Figures 3,4,5,7,8: Temperature unit is "C" instead of "C". Figure 6: Here the unit is missing in the figure (and given in the caption instead) Figure 9: Here you use "K" –> Please unify.
We changed the units to degrees Celsius throughout the paper and corrected the figure labels and captions.

---

## Author Comment (AC2) · 16 Sep 2020

Dear Michael Warscher,

Thank you for your review of our manuscript and suggestions for improvement. Please find our replies to your comments below in blue.

Best regards,
Emily Collier & Thomas Mölg

In their manuscript "BAYWRF: a convection-resolving, present-day climatological atmospheric dataset for Bavaria", the authors present a new high-resolution RCM simulation using WRF and ERA5 reanalysis data as boundary condition. They evaluate the performance for the target region of Bavaria using station observations.

**General Comments**
The manuscript is very well written and of high technical and scientific quality. It fits very well in the scope of ESSD. However, I have several issues, questions, and suggestions which at large could lead to major revisions. However, I understand that the manuscript is mainly an overview of the presented data and thereby, the amount and detail of the analyses and following content has to be limited at certain points. I would gladly leave the decision to the editors on how much of and at what detail level my suggestions should be addressed. The dataset they produced is generally very valuable for the scientific community, as well as for many users in different sectors.

The authors have chosen a single year as specific validation period. In addition, they point out in L. 101 that it is not an average year in terms of seasonal climatology (record heatwave in 2018). The chosen year might therefore not be a representative period for the RCM performance in other years. However, an extension of the evaluation period seems just limited by a missing run using the NO_NUDGE configuration. As the whole exercise is a historic / present-day reanalysis driven simulation effort, the NUDGE setup was run for the whole 30-year period anyways and would be available for additional validation years. I would highly recommend to add at least one additional year of validation (the more the better) to strengthen the results under different conditions. The validation could potentially even be done for the whole 30-years (just being limited by available observations and – by now - the missing NO_NUDGE for more than one year). I am also quite sure that an extension of the analysis would not limited by available observation data. If no additional simulations (NO_NUDGE) can be performed, the authors might think of some additional validation using the 30-year NUDGE run only.

While I really acknowledge the direct comparison to station data, the study would highly benefit from a comparison to gridded observation data sets such as REG- NIE (1 km, https://www.dwd.de/DE/leistungen/regnie/regnie.html) or HYRAS (5 km, https://www.dwd.de/DE/leistungen/hyras/hyras.html). Besides the correct representation of single stations, the real benefit of such a computationally expensive high- resolution simulation might or should be – besides the reproduction of observed station data - the resolving of spatial distributions.
To address this comment, we replaced the trend analysis with a new section evaluating the performance of BAYWRF over the whole simulation period compared with all available station data from the DWD datasets TT_TU_MN009, RF_TU_MN009, and R1_MN008 (*T*, *RH* and *PREC*) at daily timescales. We note that the *PREC* dataset is only available after 1 September 1995. We therefore also added a brief pattern correlation analysis between

simulated monthly total precipitation and the suggested REGNIE dataset for the full simulation period.

The authors point out, that they use a convection-permitting resolution of 1.5 km. However, the topic of simulating convective precipitation is not referred to again in the manuscript. This is still a very important and relevant topic, and the presented data would be ideally suited to look into this. Several questions arise and could easily be tackled. E.g. the authors show an underestimation of precipitation at some point. Could this be explained by the 1.5 km still being too coarse to resolve all or enough convective events? Are the results of the KF parameterization in the 7.5 km (D1) similar or totally different? Are sub-daily precipitation dynamics captured? Some of these questions could quite easily be investigated by comparing your results of D1 (convection parameterized) and D2 (convection resolved) to gridded precipitation products such as REGNIE and maybe even to station data.

This leads to another question regarding the resolution. While I do not question the validity and satisfying performance of the presented simulation, I would be very glad to see more about the added value of such a high resolution. This could be done by a comparison of the performance between the results of Domain D1 and Domain D2. There are no analyses in the manuscript that try to address this important question.
We agree that the question of convective characteristics is an interesting one, yet do not evaluate sub-diurnal precipitation or the added value in D2 in this manuscript for several reasons. do not evaluate sub-diurnal precipitation or the added value in D2 in this manuscript for several reasons. First, we made this dataset for (dendroclimatological) impact studies, which require kilometer-scale resolution but only daily (e.g., Dietrich et al., 2019) or even monthly temporal resolution. Given the intended application of the dataset for impact studies, we have also only made data from D2 available through the OSF repository, since users interested in climate data at ~ O(10 km) grid spacing are likely to use the ERA5 data directly.

Second, we note that there is already some consensus in the literature as to the added value of kilometer-scale grid spacing, in particular for precipitation (e.g., Ban et al., 2014; Mölg and Kaser, 2011; Prein et al., 2015) and examination of this scientific question (data interpretation) would appear to be outside of the scope of ESSD.

Finally, the full two-hourly time series of precipitation alone amounts to nearly 50 GB of data. As our project already exceeds the desired size for (free) data storage on the OSF, uploading more data to the repository is problematic – as would be presenting and evaluating data that are not available for download.

Another important issue regarding the trend analysis can be found in the specific comments.

**Specific Comments**

L. 12: I suggest to remove the reference to the project here (and at other positions in the manuscript) and state the project name solely in the acknowledgement section.
We removed the reference to the project from the abstract and the last paragraph of the introduction. Please see below for our reply about removing references elsewhere in the manuscript.

L. 32 – 38: I see that the linkage to dendroclimatological studies refers to the research project, but in my opinion, this is not needed here. You don't show any further results

regarding this topic, and the general effort and method of dynamical downscaling does not really need to be justified or explained within this manuscript.

While we agree that the introduction could be re-formulated to focus only on climate downscaling, we feel the project description and dendroecological discussion in the introduction provides an important context for why we generated these data as well as some configuration choices (e.g., the temporal resolution of the dataset). While we have removed two references to our specific project in the manuscript, we would prefer to keep some mention of this information in the introduction.

L. 49 – 59: The same as the comments above: this is in general interesting information, but not within this manuscript. The paragraph should be shortened, maybe only keep the last sentence: "High-temporal. . ."

Please see above.

L. 66 – 68: Please remove the sentence: "These data. . ." for the reason stated above.

We removed this sentence.

L. 69: While your statement here is certainly true, I would prefer a more moderate phrasing, e.g., "These data has the potential to find. . ." instead of "These data will also find. . .".

We made this change.

L. 78: Could you please give some more information on how the WRF configuration was chosen? This should then also be added to the manuscript. You state that the setup is based on Collier et al. (2019) but the study seems to be located in completely different climate and terrain conditions (East Africa). It is widely shown in literature that the performance of the chosen configuration strongly depends on the region. I understand that it is not feasible to perform a full configuration optimization ensemble, but some more information on this issue should be added.

The configuration is based on all of our previous convection resolving modelling efforts, including some mesoscale and LES simulations for the European Alps that are as-of-yet unpublished and therefore cannot be cited. The configuration was not specifically optimized for the Bavaria domain, given temporal and computational constraints. To clarify, we amended this sentence to: "The physics and dynamics options used in the simulations are based on several recent convection-permitting applications of WRF by the authors (e.g., Collier et al., 2019) but were not specifically optimized for these domains due to the computational expense of the simulations."

L. 122: It would be very interesting to see sub-daily results also for precipitation from such a simulation. By permitting convective events, this could potentially be one of the strong points of such a high-resolution simulation.

Please see our response to the general comment on this issue.

L. 123: What about all the cases where modeled precipitation > 0 but the observed precipitation = 0? These cases should somehow be analyzed too and not be neglected in the performance analysis.

Excellent point. If these events are included in the evaluation in Fig. 3, the bias evaluation is artificially improved, because WRF underestimates the magnitude of observed precipitation events and the magnitude of simulated "false" events is generally small (for example, the mean and median magnitudes are ~ 0.6 and 0.1 mm /day in WRF_NUDGE). We therefore left Figure 3 as is but added the following sentence to Section 3.1: "In addition to underestimating observed daily precipitation events (total sample size of 35,791 for all stations and record lengths), the simulations also produce false daily precipitation events,

the vast majority of which are very small in magnitude (the median value in both WRF simulations is less than 0.1 mm/day). Considering wetter days (precipitation exceeding 1 mm/day; Ban et al., 2014), the number of false events is more than ten times smaller than the number of observed events (sample sizes of 3,096 and 2,249 in WRF_NO_NUDGE and WRF_NUDGE, respectively)."

L. 128: Why did you choose two hourly WRF output? I see that the output somehow has to be confined, but hourly values would also be very valuable! Do you still have these available? I think it is fine for the manuscript to keep two hourly results, but at least for the main surface variables, it would be very useful to have hourly values as well. If they are available upon request, you could add this information to the data availability section.
The history write frequency for the WRF simulations was set to two-hourly, so hourly data were not stored and are not available. This choice was made as a compromise between temporal resolution and storage requirements, and because the dendroclimatological part of the project only requires daily resolution. At two-hourly temporal resolution, the unprocessed model output already amounts to 55 TB of storage, which represented a huge logistical challenge to store, analyze, and make available through the public repository.

We added a sentence to emphasize this point in the last paragraph of Section 2.1: "We selected this write frequency as a compromise between high-temporal resolution and the logistical challenges of storing, analyzing, and disseminating the data.

L. 147 - 157: I highly appreciate the very well investigated and documented error handling here!
Thank you!

L. 160: See comment above: why did you choose two-hourly output? Was it just to save storage or is there another reason?
Please see our reply above.

L. 169 - 170: It is very valuable that you try to compare the results to other studies (here to the work by Warscher et al. 2019), but the numbers are not really comparable here (different investigation area, nudging strategy, stations, terrain, etc.). I would either keep your statement and add an explanation regarding the differences in the analyses or remove the statement or phrase it differently ("similar but lower" is quite inexplicit).
Our intention was to contextualize our results with previous literature without going into great detail, since method comparison is outside of the scope of ESSD and the exact biases depend on many factors. We rephrased this sentence to "These values are comparable to previous high-resolution applications of WRF over Bavaria (Warscher et al., 2019)" and hope this change addresses your concern.

L. 180 - 181: To me, this is a strong hint that the used resolution is still not high enough to correctly simulate absolute convective precipitation amounts. That's one reason why it would be so valuable to analyze more than one year of data and to compare results between D1 and D2.
Given the resolution requirements for explicitly simulating the evolution of individual clouds, it is likely that some influence of moist convection is not being captured at 1.5-km grid spacing. We added to this paragraph: "The reported seasonal and mean biases in daily precipitation are consistent with a potential underestimate of deep convection and convective precipitation at 1.5-km grid spacing. Although simulated mean precipitation shows a weak grid dependency below a spacing of ~ 4 km (Langhans et al., 2012), sub-kilometer spatial

resolution is required to explicitly resolve the evolution and characteristics of clouds (e.g., Bryan et al., 2003; Craig and Dörnbrack, 2008; Prein et al., 2015).

L. 213: You clearly show that the grid-nudged run is performing better than the "free" simulation, which again leads to the question of the benefits of the simulation. This result indicates that WRF adds biases compared to the ERA5 forcing simulation when not grid-nudged to them. The DWD stations you used for your validation might even have been assimilated in ERA5 which again questions to some point the added value of the simulation. Simulation drift is a well-known issue with regional climate simulations, especially when larger domains are used, as in our simulations (e.g., Prein et al., 2015 and references therein). We contacted DWD to inquire which datasets may have been used for data assimilation in ERA5 but unfortunately have not received a reply. Nonetheless, we think it is a logical consequence (and not a particular drawback of the model or simulations) that nudging towards a dataset that assimilates some observational data produces results that agree more closely with the same, or other, observations. The benefit of the simulations is the dynamically consistent (and in D2 where no nudging occurs, physically consistent) representation of local climate at the kilometer scale.

L. 215 - 216 Remove "(the temporal resolution of data available in BAYWRF)". This is not important here.
We made this change.

L. 228: Three typos: add spaces after "WRF:"
We corrected these typos.

Sect. 3.5: Trend analysis: it is quite obvious that the trends are reproduced by the simulation when it is forced by a reanalysis product such as ERA5 (and grid-nudging is used). You could think about removing the whole section, as I do not see a value in this information. If the trends would not have been reproduced, it would be an argument that something goes wrong, but – the other way round - these results are not proving a good performance of WRF (as stated in the paragraph). You simply see the overall dynamics of the forcing (which includes assimilations of historic observations and therefore reproduces historic trends).
We removed the trend section in favour of expanded model evaluation, as suggested above.

L. 241 - 246: The paragraph falls a bit short compared to the other ones. The spatial distribution of trends could be more elaborated (if a trend section is kept). The fine scale spatial differences of trends is in the end the information that is produced by the RCM simulations (see the statements regarding trends and reanalysis above).
We removed this paragraph.

L. 261 - 263: The statement regarding grid-nudging may be true, but I do not see it as a success, as the forcing obviously includes assimilated observations (see comments above).
We intended this statement to report a result rather than a success. We attempted to clarify by re-phrasing: "Comparison of simulations for the period of September 2017 to August 2018 with and without grid-analysis nudging against extensive meteorological measurements across Bavaria showed that nudging decreased the mean deviations and increased the coefficient of determinations at the majority of sites for nearly all evaluated atmospheric variables, in particular precipitation. This approach was therefore adopted for generating the full BAYWRF dataset."

Fig. 7 c) and d): If I understand it right, the values in the legend should be reversed (wrong sign).

The difference in night-time land-surface temperature was computed as MODIS minus WRF. Somewhat unconventionally, here orange colors delineate negative values, i.e. where WRF is warmer. We realize the color choice is somewhat confusing and added an additional sentence to the figure caption: "Note that the orange and red colours in panels c and d shade areas where WRF is warmer than MODIS (MODIS minus WRF is negative) and vice versa for blues."

References

Ban, N., Schmidli, J. and Schär, C.: Evaluation of the convection-resolving regional climate modeling approach in decade-long simulations, J. Geophys. Res., 119(13), 7889–7907 [online] Available from: http://onlinelibrary.wiley.com.proxy.library.uu.nl/doi/10.1002/2014JD021478/full, 2014.

Dietrich, H., Wolf, T., Kawohl, T., Wehberg, J., Kändler, G., Mette, T., Röder, A. and Böhner, J.: Temporal and spatial high-resolution climate data from 1961 to 2100 for the German National Forest Inventory (NFI), Ann. For. Sci., doi:10.1007/s13595-018-0788-5, 2019.

Mölg, T. and Kaser, G.: A new approach to resolving climate-cryosphere relations: Downscaling climate dynamics to glacier-scale mass and energy balance without statistical scale linking, J. Geophys. Res. Atmos., 116(16), doi:10.1029/2011JD015669, 2011.

Prein, A. F., Langhans, W., Fosser, G., Ferrone, A., Ban, N., Goergen, K., Keller, M., Tölle, M., Gutjahr, O., Feser, F., Brisson, E., Kollet, S., Schmidli, J., Van Lipzig, N. P. M. and Leung, R.: A review on regional convection-permitting climate modeling: Demonstrations, prospects, and challenges, Rev. Geophys., doi:10.1002/2014RG000475, 2015.

---

## Author Response (AR2)

Dear. Dr. Carlson,

Please find our responses (blue text) to the technical corrections requested by Michael Warscher (black) below. In addition to the changes described below, we also added a sample namelist from our simulations as Appendix A, to enhance the reproducibility of our results. We hope these changes make our manuscript suitable for publication in *Earth System Science Data*.

Best regards,
Emily Collier and Thomas Mölg

The authors addressed all my issues and I now suggest the publication of the manuscript. I highly appreciate the addition of Fig. 8 which gives a nice overview of performance and spatial distribution of biases for the whole model period.

I still have two small technical corrections/suggestions:
- I would prefer a continuous (not diverging) color scale for the (continuous) R^2 values on the right side panel plots in Fig. 8. The red vs. blue indicates a kind of performance threshold which is not appropriate here for R^2.
We changed the colorscale for R2 in Fig. 8 to a continuous one.

- I now understand the scale and colours in Fig. 7 c) and d). However, I would highly recommend to just reverse the calculation (WRF minus MODIS rather than MODIS minus WRF). This way you will have positive values and red colours for a warm model bias (negative values / blues for a cold bias) which is much more intuitive for the reader.
All biases in the manuscript were computed as observations minus simulations (line 133). Although we agree that plotting WRF minus MODIS would be more intuitive in terms of contour colors, we would prefer not to change the bias calculation only for Figure 7. In addition to the clarifying sentence we added to the figure caption in the last revision, we added labels to the colorbar for Fig. 7c,d.